# The Treatment Gap in Osteoporosis

**DOI:** 10.3390/jcm10133002

**Published:** 2021-07-05

**Authors:** Nazia Ayub, Malak Faraj, Sam Ghatan, Joannes A. A. Reijers, Nicola Napoli, Ling Oei

**Affiliations:** 1Division of Endocrinology and Diabetes, University Medical Center Utrecht, 3584 CX Utrecht, The Netherlands; n.ayub@umcutrecht.nl; 2Division of Endocrinology and Diabetes, Università Campus Bio-Medico di Roma, 00128 Rome, Italy; m.faraj@unicampus.it (M.F.); N.Napoli@unicampus.it (N.N.); 3Department of Internal Medicine, Erasmus University Medical Center, PO Box 2040-Na27-24, 3015 GD Rotterdam, The Netherlands; s.ghatan@erasmusmc.nl; 4Division of Rheumatology, Leiden University Medical Center, 2333 ZA Leiden, The Netherlands; j.a.a.reijers@lumc.nl; 5Division of Bone and Mineral Diseases, Washington University, St. Louis, MO 63110, USA; 6Department of Internal Medicine, Leiden University Medical Center, 2333 ZA Leiden, The Netherlands

**Keywords:** osteoporosis, treatment gap, pharmacological treatment, epidemiology, review, fracture, glucocorticoid

## Abstract

Worldwide, there are millions of people who have been diagnosed with osteoporosis, a bone disease that increases the risk of fracture due to low bone mineral density and deterioration of bone architecture. In the US alone, there are approximately ten million men and women diagnosed with osteoporosis and this number is still growing. Diagnosis is made by measuring bone mineral density. Medications used for the treatment of osteoporosis are bisphosphonates, denosumab, raloxifene, and teriparatide. Recently, romosozumab has been added as well. In recent years, a number of advances have been made in the field of diagnostic methods and the diverse treatment options for osteoporosis. Despite these advances and a growing incidence of osteoporosis, there is a large group being left undertreated or even untreated. This group of the under/untreated has been called the treatment gap. Concerns regarding rare side effects of the medications, such as osteonecrosis of the jaw, have been reported to be one of the many causes for the treatment gap. Also, this group seems not to be sufficiently informed of the major benefits of the treatment and the diversity in treatment options. Knowledge of these could be very helpful in improving compliance and hopefully reducing the gap. In this paper, we summarize recent evidence regarding the efficacy of the various treatment options, potential side effects, and the overall benefit of treatment.

## 1. Introduction

Osteoporosis is a disease with a high prevalence, which is a major public health burden on our society. It is associated with a high incidence of fragility fractures. All osteoporotic fractures increase morbidity, with hip and vertebrae fractures also leading to increased mortality. Often osteoporotic fractures require admission to hospital, particularly hip fractures, which account for 50% of osteoporotic fracture-related hospital admissions [1]. Once in hospital, these patients are also at a high risk of developing complications, such as thrombosis (27%), urinary tract infections (12–61%), and pneumonia (7%) [1], exacerbating their health problems. Thus, the treatment of osteoporosis and prevention of fractures are vital for improving patients’ health and reducing the hospital burden.

The estimation is that ten million people in the US have osteoporosis and around 34 million are at risk. Worldwide, osteoporosis causes more than nine million fractures annually. One in two women and one in five men who are 50 years of age will have an osteoporotic fracture in their remaining lifetime. Among women, the ten-year risk of any fracture increases from 9.8% at 50 years to 21.7% at the age of 80. The International Osteoporosis Foundation (IOF) reports an estimated one million quality-adjusted life years (QALYs) were lost in 2017 due to fragility fractures in six European countries [2].

In addition, the estimated lifetime risk for 50-year-old women and men of coronary disease is 39.2% and 51.7%, respectively. Cardiovascular risk management has been commonplace for years now by lipid-lowering agents, antihypertensives and on indication glucose-lowering therapies. Recent insights suggest that patients with a high cardiovascular risk profile may also have preventive benefits from anti-osteoporotic drug treatment [3].

Osteoporotic fractures typically involve the hip, vertebrae, and the distal forearm. However, because the effect of osteoporosis on the skeleton is systemic, the associated increase in fracture risk substantially affects all skeletal sites [4]. Hip fractures are associated with a greater reduction in quality of life than all other types of fracture, with excess mortality risk. The incidence of a hip fracture increases exponentially with age in both sexes. The estimated number of hip fractures worldwide will rise from 1.7 million in 1990 to 6.3 million in 2050 due to an increasing number of elderly people in the population [5]. The prevalence of a vertebral fracture increases progressively with age in both men and women [6]. According to the European Vertebral Osteoporosis Study, one in eight women and men aged 50 years had evidence of vertebral deformity [7]. In general, the incidence of wrist fracture in men is low with no apparent increase with age [8]. Some studies showed that in white women, the incidence of wrist fractures increases between the ages of 40 and 65 and then stabilizes [9].

Recent advances in drug development for the treatment of osteoporosis over the last three decades have led to effective therapies for treating osteoporosis. Despite this, osteoporosis is vastly undertreated. Concerns about rare side effects, current comorbidities, and inadequate long-term efficacy of anti-resorptive drugs have led to an increase in the number of untreated patients, referred to as an osteoporosis treatment gap [10]. The treatment gap is considered such a major concern that multiple global health organizations, such as the ASBMR, The Gerontological Society of America, and the Fragility Fracture Network have issued global calls to tackle this crisis.

The purpose of this review is to summarize the evidence concerning the treatment gap of osteoporosis. Particularly, which groups are at risk of osteoporosis and identifying the factors underlying the treatment gap.

## 2. Definition & Diagnosis

Osteoporosis is defined by low bone density and deterioration of bone architecture, which increase the risk of fractures. The diagnosis of osteoporosis is primarily determined by measuring bone mineral density (BMD) using noninvasive dual-energy X-ray absorptiometry.

## 3. Who to Treat

Indications for treatment are mostly primary osteoporosis in the age-related variant and/or sex hormone deficiency. These groups comprise mostly postmenopausal women. Postmenopausal women are at a higher risk of fractures with twice the increased risk in comparison to men [8]. The reduction of estrogen levels in women at menopause is one of the most potent risk factors for the development of osteoporosis [11]. Treatment is generally recommended in postmenopausal women with a BMD T-score of −2.5 or less, or with a Fracture Risk Assessment Tool (FRAX) score indicating an increased risk of fracture, see also Table 1.

With the FRAX, risk factors such as age, race, alcohol use, gender, body mass index, smoking history, prior personal or parental history of fracture, use of glucocorticoids, secondary osteoporosis, rheumatoid arthritis, and femoral neck BMD measurements are included to predict the ten-year probability of hip fracture or other major osteoporotic fracture. This tool can be used in conjunction with other diagnostic tools to identify patients who are candidates for treatment. However, diagnosis and treatment of secondary osteoporosis remain challenging as this often concerns premenopausal women or younger men who are not usually targeted for routine screening for osteoporosis [12]. Management of secondary osteoporosis includes treatment of the underlying disease that causes bone loss and specific osteoporosis therapy.

Guidelines from the National Osteoporosis Foundation recommend a daily intake (from supplements and diet combined) of 1200 mg of calcium and 800–1000 IU of vitamin D, regular exercise, fall-prevention strategies, avoidance of smoking and excess alcohol intake, and the use of anti-resorptive or anabolic agents, with reassessment after two to five years for the treatment of osteoporosis [13].

## 4. Pharmacological Treatment

Currently, a broad range of pharmacological osteoporotic treatments is available to inhibit excessive bone resorption or to even increase bone formation. Figure 1 provides an overview of a selection of landmark studies per pharmacological agent throughout history [14,15,16,17,18,19,20,21,22].

Osteoporotic treatments might be associated with reductions in mortality as well as morbidity associated with osteoporotic fractures. The most commonly used anti-resorptive drugs are bisphosphonates (alendronate, risedronate, ibandronate, intravenous forms pamidronate, and zoledronate) [23]. See also Table 2 [24].

Bisphosphonates are recommended as a first-line therapeutic option for the prevention and treatment of osteoporosis in postmenopausal women and men. They inhibit bone resorption by reducing the activity and viability of osteoclasts [25]. Bisphosphonates are effective at reducing the fracture risk and increasing the bone turnover markers (BTMs) and BMD at all skeletal sites. A meta-analysis that compared alendronate with a placebo in older women with high fracture risk showed a 44% reduction in the risk of vertebral fractures, displaying a hazard ratio of 0.56 [0.46–0.67], a 40% reduction in hip fracture risk (HR, 0.60; 95% CI, 0.39 to 0.92), and a 17% reduction in nonvertebral fracture risk (HR, 0.83; 95% CI, 0.74 to 0.93) [26]. Bisphosphonates are estimated to prevent 80–5000 fragility fractures for each atypical femur fracture possibly induced by treatment [13].

The human monoclonal antibody denosumab, which is a potent anti-resorptive drug against receptor activator of NF-κB ligand (RANKL) [27], is recommended as an alternative to bisphosphonates in the treatment of postmenopausal osteoporosis. Denosumab has been shown to reduce fracture risk at all skeletal sites. Compared with a placebo, denosumab showed a 68% reduction in the risk of vertebral fractures (HR, 0.32; 95% CI, 0.26 to 0.40), a 39% reduction in the risk of hip fractures (HR, 0.61; 95% CI, 0.37 to 0.98), and a 19% reduction in the risk of nonvertebral fractures (HR, 0.81; 95% CI, 0.69 to 0.95). After ending the denosumab treatment, subsequent anti-resorptive agents like bisphosphonates should be given to prevent a rebound in bone turnover and to decrease the risk of rapid BMD loss and an increased risk of fracture [28]. In a long-term treatment (up to ten years) FREEDOM trial with denosumab, persistently low rates of new radiographic vertebral fractures (0.9% to 1.86% per year), nonvertebral fractures (0.84% to 2.55% per year), and hip fractures (0% to 0.61% per year) were observed in years four to ten, supporting its efficiency for a long-term period [19].

Raloxifene is another anti-resorptive drug used for the prevention and treatment of osteoporosis in women. It is a selective estrogen receptor modulator and is effective in reducing the risk of vertebral fractures only (for 60-mg/d: RR, 0.7; 95% CI, 0.5–0.8; for 120-mg/d: RR, 0.5; 95% CI, 0.4–0.7) [20].

Unlike the anti-resorptive drugs that reduce bone resorption, the first anabolic therapy that stimulates bone formation is teriparatide. Teriparatide is a recombinant human parathyroid hormone analog that enhances osteoblastic activity to stimulate bone formation on the bone surface. A dose of 20 ug/day of teriparatide given for up to two years was found to decrease the risk of vertebral fractures by 65% (risk ratio [RR] 0.35, 95% CI 0.22–0.55) and non-vertebral fractures by 53% compared with a placebo (0.47, 0.25–0.88) [21]. Teriparatide therapy should always be followed with an antiresorptive drug such as bisphosphonates to avoid bone density decline [29]. Abaloparatide is another novel recombinant human parathyroid hormone analog. A meta-analysis that compared abaloparatide with a placebo showed an 87% reduction in the risk of vertebral fractures (HR, 0.13; 95% CI, 0.05 to 0.38) and a 46% reduction in the risk of non-vertebral fractures (HR, 0.54; 95% CI, 0.31 to 0.96) [26].

Romosozumab is a new potential anabolic therapy that targets Wnt-B-catenin pathway inhibitors. Romosozumab is a humanized antibody that binds to and inhibits sclerostin, thereby preventing osteoblast maturation and function. In phase III clinical trials romosozumab increased BMD at the lumbar spine and total hip by 13.3% and 6.8%, respectively, and reduced the risk of vertebral fracture by 73% compared with a placebo [30].

## 5. The Growing Gap in Treatment Options

Despite the great advances in the treatment of osteoporosis, a substantial proportion of patients at high risk of fractures remain untreated, either because they are not given these medications at all, or when prescribed, the patients are not taking them [31]. Hemlund et al. have provided an extensive report estimating the treatment gaps in different European countries [32]. Epidemiological inference was made by subtracting the size of the population exceeding the osteoporosis intervention threshold calculated by FRAX by the sales numbers of anti-osteoporotic drugs corrected for non-adherence. This yielded treatment gap proportions of 25 to 85% of patients estimated to be eligible for treatment but not receiving it. This translates to a total in the EU of 12.3 million untreated individuals in 2010 [10]. An additional concern is the decrease of adherence per patient over time [33].

The most important reason for this gap is fear of rare side effects and concerns regarding the long-term efficacy of the osteoporosis treatment. The main rare side effects of concern for patients are atypical femur fractures and osteonecrosis of the jaw. Often, patients decline treatment due to these concerns but do so without taking into consideration the risk-benefit ratio of these drugs. Although these side effects are very rare and not associated with all osteoporosis drugs, patient concerns about these risks are expanding to all osteoporosis drugs.

This problem is exacerbated by the absence of clear evidence supporting the long-term efficacy of anti-resorptive drugs, particularly bisphosphonates. Long-term use (over five years) of bisphosphonates was found to be associated with increased risk of the rare side effects of atypical femur fractures and osteonecrosis of the jaw. Previous reports indicated that, unfortunately, the rates of osteoporosis medication use after hip fracture demonstrate significant decline from 40.2% in 2002 to 20.5% in 2011 in the US [34].

### 5.1. Rare Side Effects and Growing Treatment Gap

Although anti-resorptive drugs are effective in reducing the risk of fracture, concerns about rare side effects have contributed to patients’ fear of taking osteoporosis medications, leaving them at high fracture risk and increasing the osteoporosis treatment gap.

The evidence for bisphosphonates’ efficacy in reducing fracture risk comes from strong clinical trials evaluating the effect of the drug over three to five years [14,16,17,18,35]. Although these studies report relatively few side effects of bisphosphonate use, they lack information on the long-term effects. Long-term use of bisphosphonates, defined as more than three years [36], was found to be associated with rare side effects, such as osteonecrosis of the jaw [37] and atypical femoral fractures [36].

Osteonecrosis of the jaw is characterized by the exposure of the mandibular or maxillary bone. The absolute risk of bisphosphonate-associated osteonecrosis of the jaw was estimated to range from 1 in 10,000 to 1 in 100,000 (or 0.001% to 0.01%) in osteoporosis patients. Concerns about the long-term safety and efficacy of bisphosphonates have been raised due to the increased risk of the rare side effects with long-term use (over five years). The risk of osteonecrosis of the jaw in osteoporosis patients on long-term oral bisphosphonate therapy has been reported to be as high as 21 in 10,000 (or 0.21%) for patients on over four years of therapy [38]. However, data from the Fracture Intervention Trial Long-Term Extension (FLEX) showed that postmenopausal women with T scores (−2.0 to −2.5) receiving alendronate for ten years had fewer vertebral fractures than women who discontinued alendronate after five years (5.3% for placebo vs. 2.4% for alendronate; 95% CI, 0.24–0.85) [15].

Atypical femur fractures are characterized by developing fractures in the subtrochanteric region and along the femoral diaphyseal. Multiple studies have demonstrated this increased risk of atypical femur fractures among patients with bisphosphonate use; however, the significance and severity of risk range from minimal (3.2/100,000/year) [39] to more pronounced (113.1/100,000/year) [40,41]. The question of severity is key as the risk of atypical femur fractures needs to be weighed against the reduction of fracture risk when prescribing bisphosphonate treatment to patients. Recently, a large longitudinal study, using data from electronic health records in California, investigated the risk of atypical femur fracture in nearly 200,000 women using bisphosphonates. The women were followed over a ten-year period to test the effect of bisphosphonate duration on the typical fracture risk. The study found that longer bisphosphonate use is associated with atypical femur fractures, with a duration of eight years or more displaying a hazard ratio of 43.51 (95% CI, 13.70 to 138.15) and an incidence of 13.10/10,000/person yrs [42], in comparison to use for less than three months. Additionally, the authors measured the risk of osteoporotic and hip fractures due to bisphosphonate use in the same individuals. After three years of bisphosphonate use, 149 hip fractures were prevented, while only two atypical fractures occurred [42]. The authors conclude that the decreases in osteoporotic and hip fractures gained by bisphosphonate use far outweigh the increased risk of atypical fractures. The results and conclusion are in line with a previous, albeit smaller study, conducted by R.M. Dell and colleagues [40].

Also, oral forms of bisphosphonates can cause upper gastrointestinal irritation, which may include heartburn, indigestion, esophageal erosion, and esophageal ulcer.

The human monoclonal antibody denosumab is well-tolerated, but adverse effects have been observed including hypocalcemia, serious infections, skin rash, and musculoskeletal pain.

The use of the selective estrogen receptor modulator raloxifene is commonly associated with increased vasomotor symptoms. It increases the overall incidence of hot flushes by +6.3% compared to a placebo [43]. Although raloxifene reduces breast cancer risk [44], it is associated with an increased risk of venous thromboembolism (RR, 3.1; 95% CI, 1.5–6.2). There was a black-box warning about a risk of osteosarcoma associated with the recombinant human PTH hormone analog teriparatide treatment. The FDA limited the treatment duration of teriparatide to two years due to the development of osteosarcoma in rats treated with high doses of teriparatide [45]. Of note, a subsequent post-marketing surveillance study of patients treated with teriparatide has not found a causal association between osteosarcoma and the use of teriparatide in humans [46]. However, in November 2020, the FDA approved the removal of the box warming regarding osteosarcoma for a longer duration of treatment (more than two years) in patients who remain at or return to a high risk of fracture. Therefore, this should help to reduce the uncertainty relating to the use of teriparatide and the increased risk of osteosarcoma, and should be useful to both clinicians and patients as it considers the possible risks vs. potential benefits of treating osteoporosis patients at high risk of fracture. The adverse effects of teriparatide included greater rates of leg cramps and dizziness compared to a placebo [21]. The second recombinant human PTH hormone analog abaloparatide has the same box warning as teriparatide. The most common adverse events seen with the use of abaloparatide were dizziness, fatigue, headache, nausea, palpitations, and postural hypotension [47].

The anti-sclerostin antibody romosozumab might increase the risk of cardiovascular complications. Romosozumab was associated with increased adjudicated serious cardiovascular events (4.9% vs. 2.5%), cardiac ischemic events (1.8% vs. 0%), and cerebrovascular events (1.8% vs. 1.2%) compared with the control group [48].

Many guidelines for the treatment of osteoporosis recommend the treatment of postmenopausal women and men that are at high risk of fractures, especially those who have experienced a recent fracture. However, because of the side effects of osteoporotic treatment, patient-specific clinical factors should be taken into account. Due to concerns about renal toxicity, bisphosphonate should be avoided in patients whose estimated glomerular filtration rate (eGFR) <30 to 35 mL/min [28]. Nonetheless, as long as a minimum of a 15-min infusion time is maintained and the patient is well-hydrated, there has been no evidence of loss of renal function with zoledronic acid in randomized clinical trials for osteoporosis. Miller et al. reviewed the trials and reported cases of bisphosphonate-associated renal damage and found only lasting side effects in zoledronic acid-associated renal impairment, which involved the dosing regimen of zoledronic acid 4 mg once every three to four weeks in cancer patients, which is a higher dose than the 5 mg once-per-year dosage for patients with osteoporosis. Moreover, cancer patients (e.g., patients with multiple myeloma or some advanced solid tumors) are known to be at risk of renal failure from compromised kidney function and hypercalcemia. Still, in practice, some clinicians administer reduced dosages of 3 or 4 mg of zoledronate yearly in osteoporosis patients with chronic kidney insufficiency in the absence of cancer. In contrast to the bisphosphonates, denosumab may be administered to patients with CKD and those with eGFRs of ≤35 mL/min/1.73 m^2^. Yet, because denosumab can cause hypocalcemia, patients with low calcium levels should be corrected before treatment initiation. In patients who have a history of or active venous thromboembolism, raloxifene should be avoided. Patients with Paget’s disease of the bone, unexplained alkaline phosphatase elevations, prior skeletal radiotherapy, primary or metastatic bone malignancy, or hypercalcemic disorders, such as primary hyperparathyroidism, should avoid using teriparatide. Abaloparatide should be avoided in patients with pre-existing hypercalcemia and those with an underlying hypercalcemic disorder, like primary hyperparathyroidism. In summary, from the above, we see that specific patient populations merit specific treatment considerations; nonetheless, most often, an alternative treatment option is available.

### 5.2. Underestimation of the Fracture Risk and Growing Treatment Gap

Some reports found that clinical calculators including FRAX may underestimate fracture risk in over half of patients [49]. This may be due to a lack of some factors known to be associated with fracture risk, such as falls [24]. Also, FRAX has not been validated for use in patients receiving osteoporotic treatment and will underestimate fracture risk in these patients. FRAX does not correctly calculate fracture risks with ages outside the stated range of 40 to 90 years. Also, one can only enter femoral neck BMD because it has not been confirmed for use with total hip or lumbar spine BMD. Finally, it does not give recommendations on whom to treat nor on exact treatment choices.

## 6. Bisphosphonate Use, Mortality, and Vascular Calcification

The reduction in osteoporotic fractures may provide substantial benefits in terms of saving lives, reducing morbidity, and reductions in healthcare costs. Many studies have reported that bisphosphonate use may, in addition, decrease the risk of mortality [50,51,52,53,54]. However, it is unclear whether the effect is causal or merely confounded by better overall health in bisphosphonate-treated patients than in non-treated patients. Further, no clearly established bone-related mechanisms have been found to explain why greater BMD may decrease mortality independent of fracture risk. However, there is an intriguing connection between BMD and vascular calcification, which may provide a mechanism of effect on mortality. Vascular calcification shows some similarities with ossification and the frequent co-occurrence of increased vascular calcification with osteoporosis is referred to as the calcification paradox [55].

Several studies have shown an association between low BMD and high coronary arterial calcification, especially in women [56,57]; however, others have failed to replicate the finding [58]. It is thought that bisphosphonates may act to reduce arterial calcification by promoting the deposition of calcified tissues as BMD. This could be a potential benefit in cardiovascular diseases. Several meta-analyses and systematic reviews have been conducted to investigate the potential relationship. The studies conducted recently confirm that arterial calcification is reduced with bisphosphonate use. However, while some studies indicate that there is no beneficial effect on cardiovascular outcomes [59], others report a reduction in cardiovascular mortality and all-cause mortality [3]. A recent prospective cohort study involving nearly 83,000 bisphosphonate users indicates that higher treatment adherence is associated with better cardiovascular outcomes, with those in the strictest adherence displaying a hazard ratio of 0.75 [0.71–0.81] [60]. These studies provide evidence of a potential mechanism between arterial calcification and BMD, which requires further exploration. However, if valid, it could provide further justification for the intervention of osteoporotic patients and even patients with moderately compromised BMD with high cardiovascular risk with bisphosphonates.

## 7. Glucocorticoid-Induced Osteoporosis

Glucocorticoids have been in use for many decades to treat a variety of inflammatory conditions [61,62,63]. The earliest reports on the increased fracture risk associated with exposure to these drugs date from the mid-20th century [64,65,66], and currently, glucocorticoid-induced osteoporosis is the leading cause of secondary osteoporosis [67]. Bone loss already occurs in the first months of treatment with glucocorticoids in a dose-dependent manner. Although the effects of glucocorticoids on bone remodeling are manifold, they eventually lead to stimulation of osteoclast activity and suppression of osteoblasts. Another indirect effect of glucocorticoid treatment is, for example, a decrease in muscle strength, thereby elevating the fall risk and thus the fracture risk [67,68].

With the advancement of osteoporosis treatment, prevention of glucocorticoid-induced osteoporosis became possible. Prophylactically prescribing osteoporosis treatment to patients receiving glucocorticoids has been proven successful in diminishing the fracture risk, most notably of vertebral fractures [68,69,70]. Nonetheless, many reports starting in the 1990s found up to 95% of patients on glucocorticoids did not receive proper osteoporosis prophylaxis [71,72,73].

Recent years have seen a number of guidelines, which include the 2017 UK clinical guideline for the prevention and treatment of osteoporosis [74] and 2017 ACR guideline for the prevention and treatment of glucocorticoid-induced osteoporosis [75], using the framework laid down by the International Osteoporosis Foundation and the European Society of Calcified Tissues [76]. Basically, these guidelines provide a way to assess the (increased) fracture risk associated with glucocorticoid treatment, which can be used to decide whether or not to prescribe osteoporosis prophylaxis. Patients on glucocorticoids aged over 70 years, with previous fractures, or receiving ≥7.5 mg prednisolone QD for more than three months are generally expected to pass the treatment threshold [76]. These conditions appear in some of the older guidelines, as well.

Additionally, Naunton et al. found that an educational program improved adherence to guidelines and significantly increased the proportion of patients on glucocorticoids receiving prophylactic treatment [77]. Carter also reported increased compliance with the 2017 guidelines from 25 to 92% in a UK general practice following education and active identification of patients at risk of glucocorticoid-induced osteoporosis [78].

Despite the improvements in prescribing prophylaxis for glucocorticoid-induced osteoporosis following the introduction of guidelines and educational programs, some patients may still be treated suboptimally. Future research will hopefully identify the factors causing this specific treatment gap.

## 8. Future Directions

Fractures caused by osteoporosis are a major cause of morbidity and mortality, especially among elderly people. Despite advances in fracture risk assessment and osteoporosis treatment to reduce fracture risk, only a minority of patients with osteoporosis are treated. Most hip fracture patients lose the ability to live independently. Hip fractures have serious consequences in terms of reduced function and increased disability. The loss of mobility and independence, combined with admission to a nursing home following a hip fracture, is a real fear for older people. Therefore, it is important for patients with osteoporosis to understand that treating osteoporosis can reduce the risk of hip fractures, which, in turn, can reduce the risk of loss of independence and admission to nursing homes.

The rare side effects of anti-resorptive therapies have become a major concern for patients with osteoporosis and contribute to the treatment gap. Several steps can be taken to address this issue: improved education of both patients and doctors on the side effects vs. the benefits of osteoporosis drugs, increased awareness of newly-developed drugs that lack these side effects among patients and doctors, and routinely following up and re-evaluating patients, such as through fracture liaison services; also, identification of patients in high-risk groups such as the aforementioned glucocorticoid-induced osteoporosis.

Another step could be screening for atypical femoral fractures using extended femur scans by DXA to monitor patients on bisphosphonates and denosumab [79].

Hopefully, future research regarding pharmacogenomic markers will also aid in identifying patients at increased risk of atypical femur fractures, which will probably help in narrowing the treatment gap.

## 9. Conclusions

Osteoporosis-related fractures are associated with excess morbidity, mortality, and healthcare costs. Recent decades have seen a dramatic transformation in our understanding of osteoporosis epidemiology, pathophysiology, diagnosis, and treatment. The availability of multiple novel therapeutic options for fracture prevention and treatment may reduce the burden of fracture on our societies. However, the fear of rare adverse effects of osteoporotic medications and concerns regarding their long-term efficacy are increasing the osteoporosis treatment gap. This gap could be narrowed by improved awareness of both the patient and doctor on the risks of not receiving the treatment and the risks and benefits of osteoporotic treatment. Moreover, there is still an important need to find ways to improve patients’ acceptance of these effective medications and to continue developing new drugs that do not cause these side effects, with long-term efficacy. Such changes could result in a true shift of this potentially age-related disease.

## Figures and Tables

**Figure 1 jcm-10-03002-f001:**
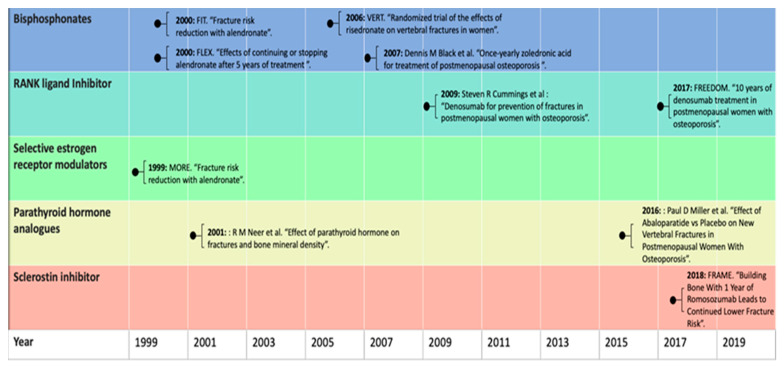
Timeline of randomized control trials in osteoporosis treatments discussed in this review.

**Table 1 jcm-10-03002-t001:** World Health Organization (WHO) diagnostic criteria for osteoporosis based on DXA bone mineral density.

**Category**	**T-Score**
Normal	−1 or higher
Osteopenia (low bone mass)	Between −1 and −2.5
Osteoporosis	−2.5 or lower

**Table 2 jcm-10-03002-t002:** Food and Drug Administration-Approved Pharmacological Interventions for Osteoporosis Treatment and Prevention.

Class/Medication	Dose	Route of Administration	Major Action	Type of Fracture Reduction	FDA Indication
**Bisphosphonate**					
**Alendronate**	10 mg daily/70 mg weekly	Oral	Anti-resorptive	Vertebral, nonvertebral, hip	Treatment and prevention
**Risedronate**	5 mg daily/35 mg weekly/150 mg monthly	Oral	Anti-resorptive	Vertebral, nonvertebral, hip	Treatment and prevention
**Ibandronate**	2.5 mg daily/150 mg monthly/3 mg every three months	Oral/intravenous	Anti-resorptive	Vertebral	Treatment and prevention
**Zoledronic acid**	5 mg yearly	Intravenous	Anti-resorptive	Vertebral, nonvertebral, hip	Treatment and prevention
**RANK Ligand inhibitor**					
**Denosumab**	60 mg every six months	Subcutaneous	Anti-resorptive	Vertebral, nonvertebral, hip	Treatment
**Selective estrogen receptor modulators**					
**Raloxifene**	60 mg daily	Oral	Anti-resorptive	Vertebral	Treatment and prevention
**Parathyroid hormone analogs**					
**Teriparatide**	20 ug daily	Subcutaneous	Osteoanabolic	Vertebral, non-vertebral	Treatment
**Abaloparatide**	80 ug daily	Subcutaneous	Osteoanabolic	Vertebral, non-vertebral	Treatment
**Sclerostin** **inhibitor**					
**Romosozumab**	210 mg monthly	Subcutaneous	Osteoanabolic/ anti-resorptive	Vertebral	Treatment

## Data Availability

Not applicable.

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
