# Peer review of "The Treatment Gap in Osteoporosis"

_jcm, 2021, doi:10.3390/jcm10133002_

Round 1

Reviewer 1 Report

The revised manuscript has improved. Please just make sure the font style and size are consistent throughout the manuscript. 

Author Response

Dear reviewer,

Thank you for your comments.

Regarding the font and size, they have been changed.

Reviewer 2 Report

Changes made to the ms are sufficient.  Accept

Author Response

Dear reviewer,

Thank you for your comments and acceptance.

We appreciate your time and effort for reviewing our manuscript.

Reviewer 3 Report

Thank you for the good and comprehensive review on the Treatment gap of osteoporosis therapy. From my point of view, the authors have addressed all relevant points.Only in the Future Directions, I think it is worthwhile to mention the efforts to increase treatment adherence via a Fracture Liaison Service.

Author Response

Dear reviewer,

Thank you for your comments.

The fracture liason service has been mentioned in line 409-412.

Do you have any further suggestions on that part?

This manuscript is a resubmission of an earlier submission. The following is a list of the peer review reports and author responses from that submission.

Round 1

Reviewer 2 Report

The review summarized the recent advances in osteoporosis treatment, treatment gap, related side effects, and their benefits in prevention of osteoporotic fracture and emphasized the gap in osteoporosis. This review provides important information for osteoporosis clinicians and fits the Journal with good writing. I recommend accept with minor revision.

Following are the minor comments:

  1. Manuscript font sizes are different. Table format needs to be refined and make it clear. The table 2 rows are not aligned.
  2. Figure 1 is not clear. If make it a figure, it is better to make in Photoshop.
  3. Line 313 sentence is not clear. Please rephrase.
  4. Line 366 is not clear. Please rephrase.

Reviewer 3 Report

This is a manuscript that examines the gap in osteoporosis treatment.  The authors describe the types of pharmacological treatments available to treat this problem.  They also state that the side effects that maybe a cause for this gap. Statistical information provided was also appropriately referenced. Overall this paper is very well written and very clear for the reader to understand.  I believe the future directions are appropriate and well stated.  Their conclusions are supported by the facts stated throughout the paper.  I only have a few minor suggestions.

  1. Why is some of the font so small.  I found it very difficult to read.
  2. Table 1 is very difficult to read d/t the small font.  Maybe it was just my computer not loading correctly?
  3. I was expecting to see some type of racial treatment gap.  Is that not the case?  I don't necessarily think you need to add it to the paper but it might be interesting in future studies.

Reviewer 4 Report

  1. Table 2 page 3 FDA approved medications does not include Romosozumab and it should include this medication.
  2. Page 3,line 75, the Flex trial with alendronate show that 10 years of bisphosphonate reduce the risk of clinical vertebral fractures. Also the FREEDOM extension trial showed that denosumab had continued benefit and no new safety signals with 10 continuous years of use.
  3. Page 6,line 241, in November 2020 the FDA softened the labeling regarding osteosarcoma with teriparatide and no longer restricts it to 2 years of use .
  4. Page 9, line 361, Future Directions:  Two important elements are  missing in your discussion which are  the fear of loss of independence and the fear of admission to a nursing home. These fears are a priority for many elderly across the world, see Quine, S. and Morrell, S. (2007), Fear of loss of independence and nursing home admission in older Australians. Health & Social Care in the Community, 15: 212-220. https://doi.org/10.1111/j.1365-2524.2006.00675.x . These 2 feared events are consequences of hip fractures and spine compression fractures. Therefore the  treatment gap should be approach by helping patients understand that treating osteoporosis can reduce the risk of hip fractures which in turn can reduce the risk of loss of independence and admission to nursing homes.